Detection of microplastics in Litopenaeus vannamei (Penaeidae) and Macrobrachium rosenbergii (Palaemonidae) in cultured pond

Reunura Thanya
Prommi Taeng On faastop@ku.ac.th
Department of Science, Faculty of Liberal Arts and Science, Kasetsart University, Kamphaeng Saen Campus , Nakhon Pathom Province , Thailand
Waiho Khor
Electronic publication date: 2022 Feb 8
Publication date: 2022
Volume: 10
Electronic Location ID: e12916
Received 2021 Oct 27; Accepted 2022 Jan 20
Copyright: ©2022 Reunura and Prommi
Copyright year: 2022
Copyright holder: Reunura and Prommi
License: This is an open access article distributed under the terms of the Creative Commons Attribution License, which permits unrestricted use, distribution, reproduction and adaptation in any medium and for any purpose provided that it is properly attributed. For attribution, the original author(s), title, publication source (PeerJ) and either DOI or URL of the article must be cited.
License URL: https://creativecommons.org/licenses/by/4.0/

Keywords: Microplastics, Freshwater prawn, Gastrointestinal tract, FTIR spectroscopy

Funding: Kasetsart University’s Graduate School This work was supported by Kasetsart University’s Graduate School Fellowship Program for the 2020 academic year to Thanya Reunura. The funders had no role in study design, data collection and analysis, decision to publish, or preparation of the manuscript.

==============================
Background

The presence of plastic particles in freshwater species is becoming a global concern owing to the potential impact on food security and human health. In this study, we investigated the presence microplastics (MPs) in two economically important freshwater species: the giant freshwater prawn (Macrobrachium rosenbergii) and the white leg shrimp (Litopenaeus vannamei) cultured in a polyculture pond in the central part of Thailand.

Methods

The gastrointestinal tract (GT) of 300 giant freshwater prawn (160 female and 140 male) and 150 white leg shrimp specimens were investigated for the presence of MPs.

Results

From the pooled samples, a total of 1,166 MP items were identified. Specifically, the GTs of female and male freshwater prawns and white leg shrimps contained an average of 33.31 ± 19.42, 33.43 ± 19.07, and 11.00 ± 4.60 MP items per individual, respectively. Further, their mean MP contents per g of gut material were 32.66 ± 5.10, 32.14 ± 4.85, and 10.28 ± 1.19 MP items, respectively. In the GT of these species, MPs with sizes in the range 500–1000 µm, were predominant, and white/transparent MPs were most prevalent (63.67%). Furthermore, regarding the morphotypes of the MPs, fibers, fragments, films, and spheres were frequently observed, with fibers shows predominance. Specifically, the proportions of fibers in the GTs of female Macrobrachium rosenbergii, male Macrobrachium rosenbergii, and Litopenaeus vannamei were 83.3, 79.91, and 46.06%, respectively. Four MP polymer types, polyethylene, polycaprolactone, polyvinyl alcohol, and acrylonitrile butadiene styrene, were also identified via FTIR spectroscopy, which further confirmed the presence of MPs in the GT of the freshwater-cultured species.

Discussion

Our findings indicated that consuming shrimps and prawns without first removing the MPs from their GTs is one of the mean by which humans get exposed to MPs. Thus, MPs in freshwater species can be passed down the food chain to humans.

Introduction

Plastic pollution is a serious issue that endangers species in aquatic ecosystems as well as human health. Microplastics (MPs), in particular, with length in the order of <five mm (NOAA, 2016), are formed as a result of the fragmentation of larger plastic debris or are purposefully designed to be that size for commercial purposes (Arthur, Baker & Bamford, 2009). Based on their origin, they can be classified as primary and secondary MPs (Masura et al., 2015). Specifically, primary MPs are those with diameter <five mm at manufacture, whereas secondary MPs are those formed from the degradation of larger pieces of plastics over time owing to processes, such as UV radiation or photo-oxidative degradation (GESAMP, 2015, Masura et al., 2015). Further, MPs have different colors, sizes, and shapes (Wright, Thompson & Galloway, 2013), and some of the most frequently observed MP shapes are the fiber, film, fragment, and granule shapes (Rocha-Santos & Duarte, 2017). Owing to these differences in size and shape, MPs have varying densities (Wright, Thompson & Galloway, 2013). The less dense ones float on the water surface, the denser ones may be suspended within the water column, while the much denser ones may sink to the bottom of the water column (Wright, Thompson & Galloway, 2013). It has also been reported that lighter particles may become denser over time as a variety of chemical substances and microorganisms adhere to their surface (Avio, Gorbi & Regoli, 2017) given that owing to their high surface-to-volume ratio as well as their hydrophilic nature, they have a high affinity for different chemical substances and microorganisms (Carbery, O’Connor & Thavamani, 2018). This increases their densities over time; thus, they eventually sink into sediments (Avio, Gorbi & Regoli, 2017).

Plastic particles can move from land-based sources to streams and rivers owing to the action of wind, rain, waste water flow, and inappropriate plastic waste disposal (Duis & Coors, 2016). Thus, owing to their small size, they are widely distributed in the environment and can enter the human body as well as other organisms via ingestion and inhalation, causing negative effects. Additionally, their small size allows them to be bioavailable for ingestion by aquatic biota, and reportedly, their bioavailability increases as their size decreases (Carbery, O’Connor & Thavamani, 2018). Reportedly, the presence of particles at all levels of the water column also increases their bioavailability to all types of aquatic biota, from filter feeders to deposit feeders, and from small primary consumers, like zooplankton, to top predators, like sharks (Avio, Gorbi & Regoli, 2017; Carbery, O’Connor & Thavamani, 2018). Thus, MP ingestion can occur directly from water or sediments, as is the case with filter feeders and deposit feeders or indirectly via transfer through food chains (Herrera et al., 2019; Wang et al., 2019). Reportedly, the accumulation of MPs in the digestive tracts of living organisms causes internal wounds and also clogs the digestive tract of these organisms, giving a false sense of hunger satiation (Wright, Thompson & Galloway, 2013; Rocha-Santos & Duarte, 2017), and this often leads to a decrease in foot intake or its complete inhibition, resulting in malnutrition and subsequently, death (Rocha-Santos & Duarte, 2017). 

Aquatic organisms, particularly fish and shellfish, are vulnerable to MP ingestion owing to the appealing color of the MPs as well as their buoyancy, which resembles that of their food. It has also been observed that MPs can enter the food chain following the consumption of contaminated seafood or food products, and this endangers human health (Wright & Kelly, 2017; Rist et al., 2018; Su et al., 2019). Generally, ingestion is regarded as the primary route by which aquatic organisms, such as fish, shellfish, and shrimp take up MPs (Sanchez, Bender & Porcher, 2014; Lusher, McHugh & Thompson, 2013; Vandermeersch et al., 2015), and reportedly, these MPs reside in the gut or intestinal tract after ingestion and are eventually eliminated (Rummel et al., 2016; Welden & Cowie, 2016). Physical damage owing to MP ingestion, including internal and/or external abrasions, ulcers, and digestive tract blockages, has also been reported (Wright, Thompson & Galloway, 2013). While some ingested contaminants may accumulate in tissues and cause internal exposure, MP exposure is more transient in nature.

Shrimps, which are bottom water column dwellers, eat small crustaceans, mud, detritus, vegetable matter, and diatoms (Smith, Dall & Moore, 1992). Given that MPs are the same size as sediments and planktonic prey items for lower trophic organisms, the possibility of their ingestion by benthic and pelagic biota, with diverse feeding behaviors, increases as the density of MPs in water increases (Wright, Thompson & Galloway, 2013). Further, the presence of MPs in shrimps in various geographical regions of the world has been previously reported in several studies (Devriese et al., 2015; Abbasi et al., 2018; Carreras-Colom et al., 2018; Akhbarizadeh, Moore & Keshavarzi, 2019; Cau et al., 2019; Curren et al., 2020; Daniel, Ashraf & Thomas, 2020; Hossain et al., 2020; Nan et al., 2020; Gurjar et al., 2021). In this study, shrimps were chosen because they play an important role in the food chain and provide food for a variety of animals, ranging from fish to humans. Further, to the best of our knowledge, no study has been conducted to investigate the presence of MPs in shrimps or prawns reared in inland water ponds, Therefore, in this study, we investigated the presence of microplastics in two freshwater-cultured species (Litopenaeus vannamei and Macrobrachium rosenbergii), which are popular foods in both domestic and international markets. 

Materials & Method

Sample collection and processing

Fresh specimens of the two freshwater organisms, the giant freshwater prawn (Macrobrachium rosenbergii) and the white leg shrimp (Litopenaeus vannamei), were obtained directly from three local shrimp polyculture ponds in the central region of Thailand using the modified seine net on October 20, 2020.

In total, 300 giant freshwater prawn (160 females and 140 males) and 150 white leg shrimp specimens were collected, after which they were kept cold in an icebox before transportation to the Zoology Laboratory of the Faculty of Liberal Arts and Science (Kasetsart University, Kamphaeng Saen Campus, Nakhon Pathom Province, Thailand). In the lab, the specimens were stored at −20 °C until analysis. Specifically, before analysis, the frozen specimens were defrosted in a metal tray and rinsed twice with deionized water (DI). Thereafter, their body weights were measured and documented, and 10 male giant freshwater prawns (×14 replicates), 10 female giant freshwater prawns (×16 replicates), and 15 white leg shrimp (×15 replicates) with identical weights were grouped to for the analysis of MP ingestion. Metal forceps and metal scissors, which were cleaned with DI after each batch of specimens were processed, were used to individually dissect the specimens on metal trays and remove their gastrointestinal tracts (GTs), respectively. The GTs were then transferred into 100-mL glass beakers and weighted.

To prevent any form of MP contamination, all the lab surfaces and glassware were thoroughly cleaned using 70% ethanol and ultrapure water before commencing the lab work. Further, to prevent MP cross-contamination between the specimens, the forceps were carefully rinsed after the removal of the GTs from each specimen. Finally, to prevent airborne MP contamination, the Petri dishes containing the GTs were immediately covered with aluminum foil. 

Hydrogen peroxide treatment

MPs were extracted from the GTs of the specimens using a 30% H2O2 solution. To break down the soft tissue, 20 mL of 30% H2O2 was added to each of the glass beakers (Avio, Gorbi & Regoli, 2015), which thereafter, were wrapped in parafilm and shaken at 150 rpm for 7 days, until all of the organic matter was digested. The blanks were run parallel to the soft tissue disintegration and scanned for the presence of MPs. Thus, no parafilm or MP particles were detected in the blanks.

Potassium formate floatation and filtration

MPs were separated from the dissolved organic matter solution via HCO2K flotation and filtration (Zhang et al., 2016). Each sample was placed in a glass separation funnel and saturated with HCO2K (99%) until the solution reached 1.6 g ml−1. Thereafter, the samples were maintained at room temperature for at least 3 h. The saturated solution allowed the less dense particles to separate, resulting in a layer of MPs floating upwards, while undissolved organic leftovers and inorganic materials sank to the bottom of the glass containers. The samples were then filtered using a nylon membrane filter (pore size, 0.45 µm; diameter, 47 mm; Whatman, Kent, UK) with a pressure filtration device. After this step, each membrane filter was placed in a clean Petri dish, covered with aluminum foil, and dried for 2 days at 50 °C in a drying cabinet.

A stereomicroscope (Leica EZ4E) was used to visually analyze each filter for the presence of MPs, which were identified based on their color and shape (Hidalgo-Ruz et al., 2012). Further, the shapes of the MPs were classified as fiber, sphere, film (thin and small layer), or fragment (part of a larger plastic item) (Su et al., 2018).

FTIR analyses of MPs found in the GT of the specimens

A Hyperion 2000 FT-IR microscope equipped with a mercury-cadmium telluride detector (Bruker Daltonik, Billerica, MA, USA) was used to manually evaluate 20 particles from the GT of the specimens at wavenumbers in the range 4000–600 cm−1, with 32 co-added scans and at a spectral resolution of four cm−1. OPUS software version 7.5 (Bruker) was used to compare the collected spectra to those in the Bruker database. Only particles with a hit quality threshold >700 were designated MPs, as previously described (Bergmann et al., 2017).

Data analysis

The MP type, size, and color were analyzed and measured for each shrimp and prawn. Pooled samples of 10 male giant freshwater prawns (14 replicates), 10 female giant freshwater prawns (16 replicates), and 150 white leg shrimp specimens (15 replicates) were used to calculate the average number of MPs per g of the GT. One-way ANOVA in combination with Tukey’s (HSD) post hoc pairwise comparisons was performed to determine significant differences in the abundance of the MPs in the shrimp and prawn species using SPSS software version 20.0 (IBM, Armonk, NY, USA). Further, to generate graphs, Microsoft Excel 2013 (Microsoft Corp., Redmond, WA, USA).

Results

Abundance of MPs

A total of 300 giant freshwater prawn (160 female and 140 male) and 150 white leg shrimp specimens were examined. Our analysis confirmed the presence of MP in both female and male giant freshwater prawns (Macrobrachium rosenbergii) as well as white leg shrimps (Litopenaeus vannamei) (Table 1).

Female and male M. rosenbergii weighed 23.71 ± 4.72 and 59.32 ± 7.64 g, respectively, while L. vannamei weighed 20.78 ± 3.99 g (Table 1). Further, a total of 533 (range, 11–74), 468 (range, 12–72), and 165 (range, 4–23) MP items were observed in the 45 replicates of pooled samples from female and male M. rosenbergii and L. vannamei, respectively, and the average number of MP items per individual for these three specimens were of 33.31 ± 19.42, 33.43 ± 19.07, and 11.00 ± 4.60, respectively.

Furthermore, the MPs in the GT of female and male M. rosenbergii and L. vannamei were 32.66 ± 5.10, 32.14 ± 4.85, and 10.28 ± 1.19 MP items/g GT, respectively (wet weight). Based on one-way ANOVA, our analysis also showed significant differences between the total number of MP items corresponding to the different specimens (F = 9.838; p = 0.000) (Table 1).

MP sizes

Figure 1 shows the size class frequency distribution of the MPs in the acid digested GTs of the specimens. The identified MPs could be grouped into four different size ranges: <250 µm, 250–500 µm, 500–1000 µm, and 1000–5000 µm, and in all the specimens, MPs corresponding to all the four size categories were observed.

Table 1 Microplastic abundance in various prawn and shrimp species.

Species	Number of individuals studied	Body weight (g)	Gastrointestinal (GT) tract weight (g)	Microplastics (MPs) item	
				Total MPs	Average MPs/ individual	Average MPs/g GT	
Litopenaeus vannamei	150
(×15 replicates)	20.78 ± 3.99	1.07 ± 1.04	165	11.00 ± 4.60a	10.28 ± 1.19	
Macrobrachium rosenbergii (female)	160
(×16
replicates)	23.71 ± 4.72	1.02 ± 1.01	533	33.31 ± 19.42b	32.66 ± 5.10	
Macrobrachium rosenbergii (male)	140 (×14 replicates)	59.32 ± 7.64	1.04 ± 1.02	468	33.43 ± 19.07b	32.14 ± 4.85	
Notes.

a Significant difference (p < 0.05).

b No significant difference (p < 0.50).

Figure 1 Microplastics in shrimp and prawns: sample preparation, digestion, and analytical processes.

The GT of female M. rosenbergii predominantly contained large-size MPs (500–1000 µm; 54%,289 items), whereas MPs with sizes in the ranges 250–500 µm, 1000–5000 µm, and <250 µm constituted 31% (165 items), 8% (42 items), and 7% (36 items) of the MPs in this species, respectively.

In the GT of male M. rosenbergii, large-size MPs (500–1000 µm) were also predominant (54%, 255 items), while MPs with sizes in the ranges <250 µm, 250–500 µm, and 1000–5000 µm constituted 23% (106 items), 15% (69 items), and 8% (38 items) of the MPs in this specimen, respectively.

Further, in the GT of L. vannamei, all four MP size categories: <250 µm, 500–1000 µm, 250–500 µm, and 1000–5000 µm, with proportions 38% (63 items), 37% (61 items), 15% (25 items), and 10% (16 items), respectively, were observed. Additionally, L. vannamei had a higher proportion of smaller MPs, possibly owing to its smaller size (i.e., 20.78 ± 3.99 g) compared with male M. rosenbergii (59.32 ± 7.64 g). Thus, MP size proportion distribution was significantly affected by the species investigated (χ2 = 124.766; df = 6; p = 0.000).

MP type and color

Fiber-, fragment-, film-, and sphere-shaped MPs, which differed in proportion between the specimens (χ2 = 116.396; df = 6; p = 0.000), were observed in the GT of the different specimens (Table 2).

Table 2 Microplastic type and color in two shrimp species.

Category of microplastics	Shrimp species	
		Litopenaeus vannamei	Macrobrachium rosenbergii (male)	Macrobrachium rosenbergii (female)	
Type (%)	Fiber	46.06	79.91	83.3	
	Fragment	45.45	16.67	16.33	
	Film	8.48	2.99	0	
	Sphere	0	0.43	0.38	
Color (%)	Black	17.58	10.26	9.38	
	Red	8.48	2.99	5.25	
	White/transparent	49.09	64.53	78.05	
	Blue	6.67	9.19	2.06	
	Yellow	17.58	12.61	5.25	
	Green	0.61	0.43	0	

Fiber MPs (83.3%) showed dominance in the GT of female M. rosenbergii, followed by fragment MPs (16.33%) and sphere MPs (0.38%). No film MPs were observed. Conversely, in the GT of male M. rosenbergii, all four MP types were observed, with fiber MPs showing predominance (79.91%) followed by fragment (16.67%), film (2.99%), and sphere (0.43%) MPs. Further, in the GT of L. vannamei, fiber also showed dominance (46.06%), followed by fragment (45.45%), and film (8.48%) MPs, while sphere MPs were absent. Figure 2 shows a variety of MPs of different shapes, colors, and sizes.

Figure 2 Microplastic size distribution in the gastrointestinal tract of shrimp.

For each species (n = 45), 45 replicates were created, with 10 individuals pooled in each replication.

The MPs observed had six distinct colors, the most prevalent of which was white (transparent) MPs, followed by black, yellow, blue, red, and green MPs. In the GT of male M. rosenbergii and L. vannamei, all six MP colors were observed, whereas in the GT of female M. rosenbergii, green MP particles were absent (Table 2). Additionally, the MP color proportions differed significantly between the specimens (χ2 =83.938; df = 10; p = 0.000).

MP polymer types

Based on FT-IR analysis, 16 of the 20 randomly selected particles were identified as plastic material, while 4 were identified as non-plastic material, and of the 16 MP particles, 13 were polyethylene (65%), while 3 others were identified as polycaprolactone, polyvinyl alcohol, and acrylonitrile-butadiene-styrene polymers (5% each) (Table 3). The FT-IR spectra further showed peaks at 2900 and 1500 cm−1, 2950 and 1250 cm−1, 3350 and 2900 cm−1, and 2900 and 1500 cm−1corresponding to polyethylene, polycaprolactone, polyvinyl alcohol, and acrylonitrile-butadiene-styrene showed peaks, respectively (Fig. 3).

Table 3 Microplastic polymers identified via FT-IR.

Description	Number	Percentage (%)	
Total particle measured (random selection)	20	100a	
Total polymer identified	16	80b	
PE (Polyethylene)	13	65c	
PCL (Polycaprolactone)	1	5c	
PVA (Polyvinyl alcohol)	1	5c	
ABS (Acrylonitrile-Butadiene-Styrene)	1	5c	
Total non-plastic particle	4	20	
Notes.

a Percentage of analyzed MP particles.

b Percentage of polymers in analyzed MP particles.

c Percentage of MP polymer type.

Figure 3 FT-IR analysis and photos of the most common forms of microplastics detected in samples ((A) polyethylene, (B) polycaprolactone, (C) polyvinyl alcohol, (D) acrylonitrile-butadiene-styrene).

Discussion

In this study, we confirmed the presence of MPs in the GTs of M. rosenbergii and L. vannamei, both of which are usually sold fresh in Thai markets, and also exported to other countries. Despite the fact that MPs are a well-known pollutants in freshwater, their presence in shrimps cultured in ponds has not been previously documented. Thus, to the best of our knowledge, this study is the first to demonstrate the presence of MPs in freshwater shrimps.

The average abundances of MPs in the GT of male and female M. rosenbergii were 32.14 ± 4.85 and 32.66 ± 5.10 items/g of GT, respectively. Further, in the GT of L. vannamei, their average abundance was 10.28 ± 1.19 items/g GT. Therefore, these species, which have high commercial value and are frequently consumed by humans, contain varied proportions of MPs, which they obtained predominantly via ingestion. This is consistent with reports of contamination in species from various climatic regions (Table 4). However, the abundances here reported are higher than those reported recently by Hossain et al. (2020), who identified a total of 39 and 33 MP items in M. monoceros and P. monodon from the coastal waters of Bangladesh, respectively, with average abundances of 3.87 ± 1.05 and 3.40  ± 1.23 MP items/g GT, respectively. In an earlier study, they observed an average value of 0.68 ± 0.55 MP/g (1.23 ± 0.99 MPs/shrimp) for brown shrimps from the Channel area and the southern part of the North Sea (Devriese et al., 2015). Further, in another previous study, MPs were observed in commercially important crustacean species collected from four different sites in the Musa Estuary and one site in the Persian Gulf. In this study, different forms of MPs were frequently detected in P. semisulcatus (Abbasi et al., 2018). A study involving the brown shrimp, Crangon crangon, from the southern North Sea and the English Channel reported similar findings; MPs were observed in the guts of 63% of the shrimps (Devriese et al., 2015). The presence of MPs was also confirmed in 36% of the Australian glass shrimp, Paratya australiensis (Family Atyidae), which is found in fresh waterbodies in eastern Australia (Nan et al., 2020). However, it is remarkable that in this study, all the specimens investigated (100%), which were both cultured in a freshwater pond, contained more ingested MPs. This could be because the shrimp meal, water, or prawn fishing gear represent a potential source of MPs that can be transferred to the cultured shrimps, thus posing a concern for aquaculture (Hanachi et al., 2019). The findings of this study indicate that shrimps and prawns cultured in ponds are not MP pollutant free.

Table 4 Studies on the presence of microplastics (MPs) in shrimp species.

Species	Location	Microplastic abundance	References	
Crangon crangon	North Sea	1.23 ± 0.99 items/individual	Devriese et al. (2015)	
Aristeus antennatus	Balearic basin, northwestern Mediterranean sea	39.2% individuals reported to have ingested MPs; Fibers dominant	Carreras-Colom et al. (2018)	
	Sardinia Island, Mediterranean Sea	1.66 ± 0.11 pieces/individual; Fragments dominant at 53%,	Cau et al. (2019)	
Penaeus semisulcatus	Northeast of Persian Gulf	0.360 pieces/g of muscle	Akhbarizadeh, Moore & Keshavarzi (2019)	
Fenneropenaeus indicus	Cochin, Kerala, India	0.04 ± 0.07 pieces/individual; Fibers dominant (83%)	Daniel, Ashraf & Thomas (2020)	
Penaeus monodon	Northern Bay of Bengal, Bangladesh	6.60 ± 0.2 pieces/g of gastrointestinal tract; Filaments dominant (57%)	Hossain et al. (2020)	
Metapenaeus monoceros	Northern Bay of Bengal, Bangladesh	3.87 ± 1.05 pieces/g of gastrointestinal tract; Filaments dominant (58%)	Hossain et al. (2020)	
Litopenaeus vannamei	Malaysia	20.8 ± 3.57/g w.w.; Film dominant (97.9%)	Curren et al. (2020)	
	Ecuador	13.4 ± 1.42/g w.w.; Film dominant (93%)	Curren et al. (2020)	
Paratya australiensis	Australia	0.52  ± 0.55 items/individual (24  ± 31 items/g); Fibers dominant (58.3–100.0%).	Nan et al. (2020)	
Metapenaeus monoceros	North eastern Arabian Sea	7.23 ± 2.63 MPs/individual; Fiber dominant	Gurjar et al. (2021)	
Parapeneopsis stylifera	North eastern Arabian Sea	5.36 ± 2.81 MPs/individual; Fiber dominant	Gurjar et al. (2021)	
Penaeus indicus	North eastern Arabian Sea	7.40 ± 2.60 MPs./individual; Fiber dominant	Gurjar et al. (2021)	
Litopenaeus vannamei	Thailand	11.00 ± 4.60 items/individual; Fiber dominant (46.06%).	This study	
Macrobrachium rosenbergii	Thailand	33.43 ± 19.07 items/individual; Fiber dominant (79.91% in male M. rosenbergii).
33.31 ± 19.42 items/individual; Fiber dominant (83.3% in female M. rosenbergii).	This study	

Based on size, in this study, we classified MPs under four different size groups, i.e., <250 µm, 250–500 µm, 500–1000 µm, and 1000–5000 µm (Fig. 1). Thus, we observed that MPs, with sizes in the range 500–1000 µm, were most abundant in the examined specimens. In a previous study, it was observes that shrimps inhabiting shallow water habitats of the Channel area of the Southern North Sea contain MPs with sizes in the range 200–1000 µm in their GTs (Devriese et al., 2015). In this present study, the GT of female and male M. rosenbergii predominantly contained large-size (500–1000 μm) MPs (54%), while the GT of L. vannamei showed a lower proportion of large size MPs (37%). This observation can be attributed to the fact that L. vannamei has a smaller mouth aperture than M. rosenbergii. Thus, its GT was predominated by smaller MPs (Gurjar et al., 2021). Hossain et al. (2020) found that 70% of the MPs in the GT of tiger shrimps are larger-size fractions (1–5 mm) compared with those in the GT of brown shrimps, in whose GT, smaller MPs (<1000 µm) are predominant (83%). Thus, the sizes of the ingested MPs varies according to species and sampling location. Specifically, Carreras-Colom et al. (2018) observed that Aristeus antennatus, a deep-water shrimp collected from the northwestern Mediterranean Sea, has 13 potential plastic items in its stomach, with fibers having length and widths in the ranges 1.9–26.7 mm (median 6.6 mm) and 0.012–0.032 mm, respectively.

In this study, we observed that the GT of the pooled specimens predominantly consisted of fiber MPs (69.67%), followed by fragment, film, and sphere MPs. Additionally, fiber MPs were observed in the GT of 83.3% of female M. rosenbergii, 79.91% of male M. rosenbergii, and 46.06% of L. vannamei. This high proportion of fiber MPs in the GT of the specimens could be attributed to intense shrimp feeding as well as the anthropogenic activities in the study area. Reportedly, fibers constitute the most abundant type MP pollution in marine waters worldwide (Walkinshaw et al., 2020). Our findings and observations are consistent with those of previous studies involving decapod crustaceans (Murray & Cowie, 2011), blue mussels (De Witte et al., 2014), and brown shrimps (Devriese et al., 2015), which showed overall fiber MP predominance in the guts of these species. Hossain et al. (2020) identified fiber MPs as predominant in P. monodon (57%), followed by particles (29%) and fragments (14%), while brown shrimps, were predominated by MP particles (42%), followed by fibers (32%), and fragments (26%). Further, Nan et al. (2020) reported that fiber-shaped MPs are the most frequently observed MPs in the GT of shrimps collected from Australian waters, and in the gut of the deep water shrimp, A. antennatus, Carreras-Colom et al. (2018) noted the absence of film-like MP particles. In another study involving brown shrimps from the Channel area and the southern part of the North Sea, MP contamination was found to be predominantly caused by synthetic fibers (96.5%). The plastic and film granules were relatively small-sized (20–100 mm) (Devriese et al., 2015). Further, Abbasi et al. (2018) reported that almost all the MPs ingested by prawns netted from the Musa Estuary in the Persian Gulf were filamentous fragments, containing single fibers of varying sizes. The results of this study corroborate those of previous studies given that fibers were the most frequently observed form of MP in the GT of the pond-cultured specimens investigated in this study. Furthermore, in this study, fragment MPs were found to be abundant (16–45%) in both shrimps and prawns. This is consistent with observations made by Devriese et al. (2015) and Murray & Cowie (2011) regarding brown shrimps and decapod crustaceans, respectively.

Additionally, in this study, white (transparent), black, yellow, blue, red, and green colored MPs were observed in the GT of all the specimens, with the white (transparent) MPs showing predominance (63.67% of all the MPs). Generally, MP particles can be consumed directly owing to their morphological similarity to natural food items, or indirectly owing to their adherence to food particles, and given that they are visually similar to the natural foods and preys of shrimps, their bioavailability to these organisms is relatively high (Ory et al., 2017). Similar MP color patterns have been observed in tiger and brown shrimps collected from Bangladesh waters. In this previous study, black MPs were most frequently observed, followed by transparent (white), green, blue, and red MPs. In the gut of P. monodon’s, MPs of five different colors have been observed (black, white, green, blue, and red with relative abundances of 48, 33, 11, 6, and 2%, respectively); however, M. monoceros showed a slightly different MP color pattern colors (Hossain et al., 2020). In shrimps from Australian waters, blue colored MPs were most frequently observed (90%) (Nan et al., 2020). Further, in the deep-water shrimp, A. antennatus, five different fiber colors (transparent, blue, black, red, and green) have been observed with no particular color showing predominance (Carreras-Colom et al., 2018). Further, Abbasi et al. (2018) reported that the gut of P. semisulcatus predominantly contains black or grey MP filaments (71%). It has also been reported that the brown shrimp, C. crangon, primarily ingests yellow-greenish MPs (50%) followed by purple-blue (43%), translucent (15%), and orange (12%) fiber MPs, with a small fraction of transparent (8%) and pink (2%) fibers. Possibly, translucent fibers result from colored fibers owing to exposure to acids during the acid treatment process (Devriese et al., 2015). Similar to our findings, the presence of MPs of various colors in various pelagic and demersal fishes has also been confirmed (Boerger et al., 2010; Lusher, McHugh & Thompson, 2013; Bellas et al., 2016; Ory et al., 2017; Gurjar et al., 2021).

Based on FT-IR, we identified four types of MP polymers (polyethylene, polycaprolactone, polyvinyl alcohol, and acrylonitrile-butadiene-styrene) in the GT of the examined specimens. These polymers are widely used in the manufacture of fishing equipment and ropes as well as food packaging and clothing materials (Cheung, Lui & Fok, 2018; Wang et al., 2019). Similar to our findings, Gurjar et al. (2021) reported the presence of polyethylene, polypropylene, polyamide, nylon, polyester, and polyethylene terephthalate in the GT of M. monoceros, P. stylizer, and P. indices. Further, in P. australiensis collected in Australia, 11 different types of MP polymers were observed, with rayon and polyester showing predominance (22.6 and 7.5%, respectively) (Nan et al., 2020). Polymer analysis in a recent study by Gurjar et al. (2021) confirmed that laundry and domestic wastewater, fishing gear, and food packaging materials could be the primary source of MP pollution in the study area. Therefore, in this study, the MPs in the GTs of shrimps and prawns, which is a serious issue with respect to aquaculture, possibly originated from shrimp meal, water, or prawn fishing gear (Hanachi et al., 2019).

MPs have been observed in aquatic organisms, raising concerns regarding their effect on human health (Sharma & Chatterjee, 2017; Smith et al., 2018). Further, it has been observed that the consumption of MPs by shrimps results in digestive organ damage as well as a decrease in growth and reproductive output. Usually, shrimps are peeled to get rid of the head and shell before consumption. However, given that their GTs are not always entirely eliminated during their preparation, the MPs in their intestines could be passed to humans following consumption. This is a route of human exposure to MPs that is frequently considered in terms of human health and food security (Cox et al., 2019). Lassen et al. (2015) indicated that ingested MP from toothpaste can be absorbed by the human GT, and a recent study confirmed the presence of MPs in human colectomy specimens (Ibrahim et al., 2020). However, there is no published research on the destiny of MPs resulting from human consumption of shrimps and prawns. Thus, further research is needed to determine the retention and impact of MPs from shrimps and prawns on human health.

Conclusion

In this study, our findings confirmed the presence of MPs of various shapes, sizes, and colors in shrimps and prawns harvested from a freshwater ponds in the central region of Thailand. Possibly, the MPs originated from shrimp meal, water, or prawn fishing gear (possible primary sources). Additionally, the use of FT-IR analysis to characterized the MPs observed in the examined species further enhanced the reliability of our results. By removing the intestine completely before cooking and eating, MP contamination in the edible portion of shrimps can be attenuated. Therefore, the health risks associated with MP-contaminated shrimps can be mitigated to some extent. Nevertheless, further studies are needed to investigate the accumulation of plastic debris on other edible parts of shrimps. Further, it is also necessary to clarify the potential for pollutant transfer to higher trophic levels and also investigate potential measures that can be taken to protect aquatic species from plastic pollution.

Supplemental Information

Supplemental Information 1 Raw data of microplastics in Litopenaeus vannamei and Macrobrachium rosenbergii

Click here for additional data file.

Supplemental Information 2 Raw data of Fig. 2

Click here for additional data file.

Additional Information and Declarations

Competing Interests

Author Contributions

Data Availability

The authors declare there are no competing interests.

Thanya Reunura and Taeng On Prommi conceived and designed the experiments, performed the experiments, analyzed the data, prepared figures and/or tables, authored or reviewed drafts of the paper, and approved the final draft.

The following information was supplied regarding data availability:

The raw measurements are available in the Figs. 1–3 and Tables 1–3, and the Supplementary File.

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
