# Peer review of "Detection of microplastics in Litopenaeus vannamei (Penaeidae) and Macrobrachium rosenbergii (Palaemonidae) in cultured pond"

_PeerJ, doi:10.7717/peerj.12916_

## Round 0.1 · original submission · Minor Revisions

As outlined by the reviewers, there are still several changes that need to be addressed before the manuscript can be accepted. Please kindly prepare a rebuttal/revision accordingly.

·

Basic reporting

In this work, the authors investigated the occurrence of microplastics in Litopenaeus vannamei (Penaeidae) and Macrobrachium rosenbergii (Palaemonidae) in cultured ponds of Thailand. Incidence of microplastics in two economically important shrimp and prawn species was examined. A good attempt but there are many works carried out like this in recent times.

Experimental design

The authors have reported the MPs in the gut of prawn and shrimp of polyculture pond without emphasizing on the source profiling incidence in other body parts. Methodology, results, discussion and conclusion is ambiguous and instead of male and female prawn, incidence in different sizes could be done.

Validity of the findings

English should be improved; several grammatical mistakes have to be taken care of. Flow can be improved. Discussion has to be improved heavily. Refer more studies regarding the abundance of MPs in aquacultured organisms and pond water as very few works are cited.

Additional comments

Specific comments:
1) L 33-35, 49, 51, 156, 165 so on: Please improve clarity of writing, the entire manuscript have to be re written.
2) L 47: widely accepted size range of microplastics is 1 micron to 5 millimeters (please refer Mallik, A., Xavier, K.M., Naidu, B.C. and Nayak, B.B., 2021. Ecotoxicological and physiological risks of microplastics on fish and their possible mitigation measures. Science of The Total Environment, 779).
3) L70 : State proper reason why they are more prevalent in aquatic organisms
4) L97 : expand GT Shrimp
5) L124: Both the times it is mentioned female freshwater prawns. Please correct
6) L187: please mention the processing condition of shrimp and prawns of Thai supermarkets and fresh markets
7) L159-160 – Refer mouth sizes of vannamei and rosenbergii. Please elaborate what could be the reason for significant variation in abundance, sizes, colours of MPs in these both
8) L223: try to elaborate effects of consuming MPs (please refer Ecotoxicological and physiological risks of microplastics on fish and their possible mitigation measures. Science of The Total Environment, 779 and Microplastics in shrimps: a study from the trawling grounds of north eastern part of Arabian Sea Environmental Science and Pollution Research 28: 48494–48504 http://doi.org/10.1007/s11356-021-14121-z)
9) L215- Very few MPs were subjected to chemical characterization as more than 3000 MPs were reported
10) L245- irrelevant and incorrect statement
11) L246: conclusion part doesn’t reflect any of the findings and moreover very ambiguous

Reviewer 2 ·

Basic reporting

The manuscript describes the novel findings of the presence of MPs in the giant freshwater prawn (Macrobrachium rosenbergii) and white leg shrimp (Litopenaeus vannamei) from culture farms in Thailand. As the importance of MPs in aquaculture products (especially in crustacean products) are underestimated, I think this manuscript is worthy to be published in PeerJ. I have several concerns as belows:

Experimental design

1. The authors have described that the shrimp and prawn samples were collected from the polyculture pond. Then, are the collected samples were from single polyculture pond? or combined from more than two pond? Needs to be clarified.
2. If available, please describe the information on the plastic devices and conditions of the sample-collected pond.

Validity of the findings

1. Based on the manuscript of the authors, the shrimp and prawn samples were collected from culturing farms. However, in the discussion section, the authors have stated that 'It's the first study to demonstrate the presence of microplastics in commercially marketed shrimp and prawns.' (Line Nos 190-191). Which is correct? Needs to be clarified.
2. If the two target shrimp and prawn species were collected from same culture farm or pond, the authors may need to discuss why the majority of microplastics found in gastrointestinal tracts of the two species were differed in this study. Please add some discussion in the section.

Additional comments

Table 2 and Table 3 those attached in the end of manuscript should be rearranged.
The quality of Figure 1 needs to be improved.

Reviewer 3 ·

Basic reporting

This study is on microplastics- a widespread pollutant in the 21st century. The topic is very relevant, with multiple implications in aquatic ecosystems.

Experimental design

The topic is well introduced but short, so the authors can consider lengthening the introduction. The section: contamination control is missing from the methods. It would be good to elaborate on the specific measures taken to reduce surrounding contamination of mps during sampling processing.

Validity of the findings

no comment, the results are valid

Additional comments

I have some specific comments:

Line 46- microplastics are not only from 1-5 mm, they are pieces smaller than 5mm, kindly make the change.

Line 50- please include an example size of microbeads to elaborate on the point

Line 55- kindly change “inappropriate” to “improper”

Line 61- please include some examples of the effects of mp ingestion in the organisms

Line 62- abrupt change of topic, kindly include linking statements to the previous paragraph.

Line 65- include some relevant examples of pollution/deterioration etc etc. more elaboration can be included in this paragraph

Line 71- what do you mean by “consumers”?

Line 74- please elaborate on the statement that food can be a source of mps in humans

From the results section: kindly change all mentions of general name of the prawn to the scientific name for standardisation purposes

The authors can add a table in the discussion comparing the abundances of mps in other form of shrimp around the world

---

## Round 0.2 · Minor Revisions

I applaud the authors for following through the comments and suggestions of the reviewers. The flow and information in the current version of the manuscript are well-articulated. However, I would like to suggest the authors to seek professional language assistance to ensure language consistency. The whole manuscript should be proofread again.

---

## Round 0.3 · Minor Revisions

The framework and results of the study are well structured. However, I sincerely would like to request the authors to seek professional language editing service as the manuscript could benefit from proofreading and rephrasing.

---

## Round 0.4 · accepted · Accept

The revised version of the manuscript improved tremendously after proofreading.